# Assessment of Physical and Sensory Attributes of Date-Based Energy Drink Treated with Ultrasonication: Modelling Changes during Storage and Predicting Shelf Life

Mohammad Fikry [1,2], Yus Aniza Yusof [2,3,*], Alhussein M. Al-Awaadh [4], Syahrul Anis Hazwani Mohd Baroyi [2], Nashratul Shera Mohamad Ghazali [2,3], Kazunori Kadota [5], Shuhaimi Mustafa [2,6], Hazizi Abu Saad [7], Nor Nadiah Abdul Karim Shah [2,3] and Saleh Al-Ghamdi [4]

1   Department of Agricultural and Biosystems Engineering, Faculty of Agriculture, Benha University, Toukh 13736, Egypt; moh.eltahlawy@fagr.bu.edu.eg
2   Laboratory of Halal Science Research, Halal Products Research Institute, Universiti Putra Malaysia, Serdang 43400, Selangor, Malaysia
3   Department of Process and Food Engineering, Faculty of Engineering, Universiti Putra Malaysia, Serdang 43400, Selangor, Malaysia
4   Department of Agricultural Engineering, King Saud University, P.O. Box 2460, Riyadh 11451, Saudi Arabia
5   Department of Formulation Design and Pharmaceutical Technology, Faculty of Pharmacy, Osaka Medical and Pharmaceutical University, 4-20-1 Nasahara, Takatsuki 569-1094, Osaka, Japan
6   Department of Microbiology, Faculty of Biotechnology and Biomolecular Sciences, Universiti Putra Malaysia, Serdang 43400, Selangor, Malaysia
7   Department of Nutrition, Faculty of Medicine and Health Sciences, Universiti Putra Malaysia, Serdang 43400, Selangor, Malaysia
*   Correspondence: yus.aniza@upm.edu.my

**Abstract:** Sonication is a relatively new and eco-friendly method used to extend the shelf life of food products. This study aimed to investigate the effects of ultrasonication and thermal treatments on the physical and sensory properties of an energy drink made from dates during cold storage at 4 °C. The study compared the effects of ultrasonication for 20, 30, and 40 min at 50% amplitude with thermal treatment at 90 °C for 5 min, aiming to model the changes in properties of processed drinks over time and predict their shelf life by integrating quality attributes. The results showed that total soluble solids (TSS) and electrical conductivity (EC) were not affected by cold storage and did not differ significantly between sonicated, thermally processed, and untreated samples. However, significant differences in pH; $L^*$, $a^*$, and $b^*$ values; Chroma; and sensory attributes were detected among the sonicated, thermally processed, and untreated samples. The sensory properties of the sonicated samples for 30 and 40 min and the thermally processed samples remained acceptable for up to 21 days. The study also found a positive correlation between the pH and the sweetness of the drink, as well as between the $L^*$ value and the appearance of the drink. Based on these findings, the zero-order model was able to accurately describe the real values of pH, colour characteristics, and sensory properties. Furthermore, the predicted shelf life of the drink sonicated for 40 min was longer than that of the control and thermally processed drinks, based on the colour change and pH of the drink. These results could be beneficial for beverage manufacturers seeking to control the quality properties of their products during processing and storage.

**Keywords:** ultrasonic; palm date; energy drink; quality and sensory properties; modelling

## 1. Introduction

The *Phoenix dactylifera* L., commonly known as the palm date tree, is a widely planted tree native to the Afro-Asian dry band extending from North Africa to the Middle East. As reported by the FAO [1], the world's palm date production in 2020 was approximately 9.5 million tons per year. According to Hasnaoui, et al. [2], Chandrasekaran and Bahkali [3],

the date palm fruits have a significant proportion of various nutrients such as carbohydrates (in the form of total sugars) which range from 44–88%, as well as small amounts of fat (0.2–0.5%), 15 different minerals and salts, protein (2.3–5.6%), vitamins, and a considerable amount of dietary fibre (ranging from 6.4–11.5%). In the dates industry, the cleaning and sorting stages play a crucial role in enhancing date fruit marketing. During the sorting process, the fruit is classified into different categories, with first- and second-grade dates typically consumed at the semi-ripe and fully ripe stages. Unfortunately, due to appearance defects, the third-grade dates are usually discarded or used as animal feed, resulting in environmental problems and significant economic losses to the industry. However, this date by-product can be processed into several derivative products, including date powder, honey, jam, vinegar, pastes, and date juice concentrates such as spreads, syrups, and liquid sugar. These products can be consumed directly or incorporated as food ingredients in products such as bakery and confectionary items.

To maintain the quality and safety of derivative products such as date drinks during manufacturing, it is important to preserve them adequately. Traditional methods such as sterilization and pasteurization are commonly used for this purpose. Previously, Kulkarni, et al. [4], Mtaoua, et al. [5], and Abbès, et al. [6] used heat treatment for processing palm date juice to inactivate the enzyme and extend its shelf life. While these thermal treatments can extend the shelf life of products, they are costly due to their high energy consumption and potential damage to nutritious elements like vitamins, sugar caramelization, or other quality changes that make them unappealing to consumers [7]. To overcome these issues, innovative and environmentally friendly techniques could be implemented to extend the shelf life of these products without relying on thermal treatments that compromise product quality.

Various non-thermal methods, such as pulsed electric field technology, high hydrostatic pressure, UV-C irradiation, and ultrasound, are utilized to extend the shelf life of liquid food products during storage. Ultrasound technology in particular is considered safe, environmentally friendly, and non-toxic compared to other innovative techniques [8,9]. As a result, several studies have explored the effects of non-thermal treatments on the microbial activity, physical characteristics, and chemical characteristics of various liquid food products, including date syrup [10], date juice [11], date vinegar [12], tomato juice [13], orange juice [14], strawberry juice [9,15], and grapefruit juice [16].

Conversely, changes in the quality attributes of drinks during their shelf life can lead to the loss of important nutritional elements and reduced consumer preference. As a result, ensuring food safety is critical during product development, as it ensures that the product is of acceptable quality until consumption [17]. One common approach to assessing the quality and stability of a drink during storage is by determining its shelf life. The maximum time that a food product can be stored under specific environmental conditions without any noticeable degradation in its quality or acceptability to customers is referred to as the shelf life [18], or the period of time before the product is no longer fit for human consumption [19]. Estimating shelf life is a complicated process since it depends on various parameters, such as the food itself, storage conditions, packaging, and manufacturing processes. Therefore, to predict the deterioration of food quality, quality indices need to be determined experimentally [20].

Using mathematical models is an efficient way to assess quality indices during the various stages of processing, packaging, and storage. Zero- and first-order models are commonly used equations to describe the degradation or formation of quality attributes during storage [21,22]. In the past, predictive models were created to estimate the shelf life of various products [21,23,24] based on quality indices and consumer preference. Despite the production and characterization of date fruit drinks [25,26], there are currently no scientific data available on the effects of ultrasound treatment on the properties of energy drinks made from dates.

The objective of this study was to use ultrasound technology as an eco-friendly approach to treat energy drinks made from dates. Specifically, the study aimed to (1) evaluate

the effect of ultrasound and thermal treatments on the physical and sensory attributes of the drink during cold storage, (2) develop mathematical models to track changes in the quality characteristics of the drink during cold storage, and (3) estimate the shelf life of the processed drink by integrating quality indices and consumer preference.

## 2. Material and Methods

### 2.1. Preparation of the Energy Drink from Dates

Third-grade dates (Khalas cultivar), which have appearance defects, were obtained from a local market in Malaysia because of their low prices, low marketing value, and consumer disfavor. The kernels were manually removed, then the flesh was minced using a mechanical chopper (ADE, KA1801, Hamburg, Germany). In order to prepare the energy drink from dates with a TSS of 17°brix, 4 L of filtered water at 70 °C was carefully mixed with 1 kg of date paste using a mechanical blender (Gpi, MT50, Gouda, The Netherlands) for a period of 15 min (Figure 1). Afterward, the prepared drink was divided into different groups for immediate sonication and thermal treatments.

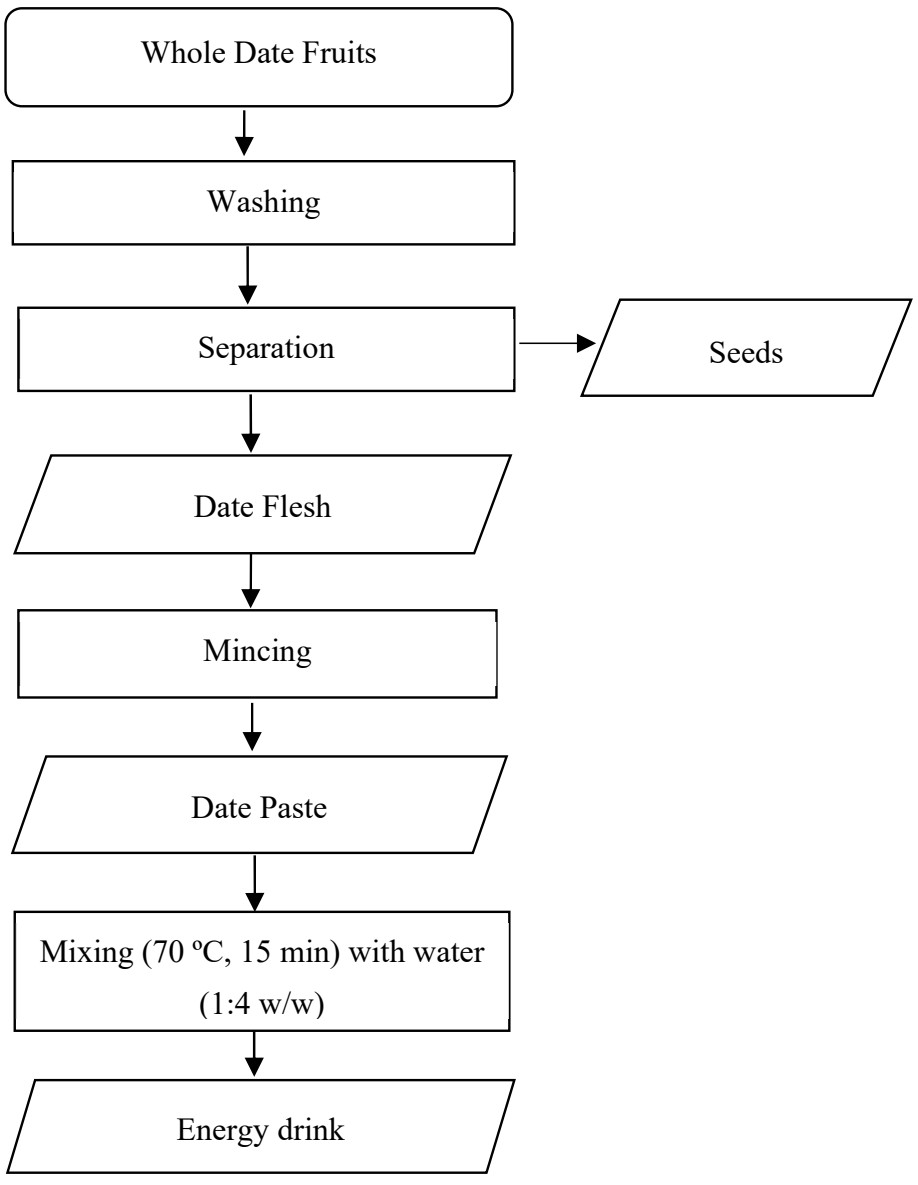

**Figure 1.** Production flow chart of the energy drink from dates.

## 2.2. Ultrasonication Process

A 500 W-ultrasonic instrument (Q500 Sonicator, QSONICA, Newtown, CT, USA) linked with a probe of 13 mm diameter and a double-jacket vessel was used. The vessel was filled with 300 mL of fresh date drink at 23.5 °C and agitated with a magnetic stirrer. The probe was immersed in the drink up to 2 cm. Samples were treated at a constant frequency of 20 kHz. The ultrasonic device was adjusted at 50% amplitude for three different treatment times (20, 30, and 40 min). Once the ultrasonic device was switched on, the temperature of the drink was raised to 56.1 ± 2 °C. The temperature of the drink was controlled by surrounding the drink-containing vessel with dry ice. The upper temperature throughout the sonication process was 55.6 ± 0.5 °C. The treated drink was cooled down to room temperature by moving it into an ice bath.

## 2.3. Thermal Treatment

According to the method previously used by Vollmer, et al. [27], 100 mL of the energy drink from dates was transferred to a glass container and placed in a water bath maintained at 90 ± 2 °C. The temperature at the core inside the water bath was detected using a K-type thermocouple (HI-93510, Hanna Equipment India, Mumbai, India). After the core reached the target temperature, the treatment period of 5 min began. Afterward, an ice bath (~2 °C) was used to cool the container.

## 2.4. Storage Study

The control, sonicated, and thermally processed drink samples were filled in low-density polyethylene bottles and then stored in a refrigerator at 4 °C. In order to perform a physical and sensory analysis of the samples, samplings at intervals of three days in the first week and then seven days for a total of 28 days were regularly conducted in triplicates.

## 2.5. Evaluation of Quality Properties

### 2.5.1. Measuring Total Soluble Solids (TSS), Electrical Conductivity (EC), and pH

A digital handheld refractometer (PAL-α, Atago Co., Ltd., Tokyo, Japan) was utilized to measure the TSS (°Brix) of the drink samples. In addition, the EC of the drink samples was also measured using a calibrated electrical conductivity meter (Cole-Parmer Instrument, Vernon Hills, IL, USA). A benchtop pH meter (Howell medical supply, Sartorius PB-10, Imus City Cavite, Philippines) was used to determine the pH of the drink samples. Before commencing, the pH meter was standardised using two buffer solutions (7 and 4.01 pH). The experiment was performed in triplicate, and the outcomes are presented as mean values.

### 2.5.2. Determination of Colour

A handheld colour reader (CR-14, Konica Minolta, Tokyo, Japan) was utilized to determine the colour parameters of the drink. Initially, the instrument (65°/0° geometry, D25 optical sensor, 10° observer) was calibrated using white ($L$ = 92.8; $a$ = −0.8, $b$ = 0.1) and black reference tiles. Colour values were expressed as a CIE $L^*a^*b^*$ system where the brightness coordinate $L^*$ and chromaticity coordinates $a^*$ and $b^*$ show lightness, redness/greenness, and yellowness/blueness, respectively [21]. All the measurements were recorded in triplicate, and mean values were reported. Subsequently, Chroma ($C^*$), which reflects colour intensity, was calculated by the following formula (Equation (1)), while the browning index (BI), which serves as the purity of brown colour, was calculated using Equation (2), according to Manzoor, et al. [28], Fikry, et al. [29], Fikry, et al. [30]. The total colour difference, the $\Delta E$, was determined, as shown in Equation (4).

$$C^* = \left(a^{*2} + b^{*2}\right)^{1/2} \tag{1}$$

$$\text{BI} = \frac{[100\,(x - 0.31)]}{0.17} \tag{2}$$

where:

$$x = \frac{a^* + 1.75\,L^*}{5.645\,L^* + a^* - 3.012\,b^*} \tag{3}$$

$$\Delta E = \sqrt{(\Delta L)^2 + (\Delta a)^2 + (\Delta b)^2} \tag{4}$$

where $\Delta L$, $\Delta a$, and $\Delta b$ are the differences between initial value and the measured value of colour components at different storage times.

2.5.3. Sensory Analysis Protocol

First, the sensory evaluation protocol of the energy drink from dates was permitted by the Ethics Committee. A nine-point hedonic scale was utilized for sensory evaluation of the control, sonicated, and thermally processed drink samples. Twenty-four semi-trained panellists performed the sensory evaluation. The evaluators were provided with information regarding the sensory characteristics of the drink, such as the appearance, odour, sweetness, and overall preference. Afterward, the drink samples were served in odourless paper cups coded with 3-digit numbers. Water rinsing between samples was applied to remove the previous samples' tastes. The sensory evaluation was conducted in a room at 25 °C and 60% relative humidity [28]. All the determinations were recorded in triplicate, and the outcomes were summarized as average values.

*2.6. Kinetics Modelling of the Quality Changes through the Storage Period*

The kinetics modelling of the quality changes was explored according to the procedure previously suggested by Fikry, et al. [22]. In order to define the reaction order of the changes in pH, colour, and sensory features, the actual data were fitted to Equation (5). Zero- and first-order formulas (Equations (6) and (7)) resulted from Equation (5) and are usually employed for portraying most reactions that indicated the degradation of food quality [29].

$$-\frac{dC}{dt} = kC^n \tag{5}$$

$$C = C_o \pm k\,t \tag{6}$$

$$\ln C = \ln C_o \pm k\,t \tag{7}$$

where $C$ and $C_o$ represent the determined property value at any time and the initial value, respectively, $t$ denotes the storage time, $k$ refers to the reaction rate constant (day$^{-1}$), and $n$ is the reaction order of the changes. The signs (+) and (−) reveal the development and degradation of the quality attribute, respectively.

*2.7. Prediction of Shelf Life of the Drink*

A least-squares fitting procedure was utilized to determine the kinetic orders by the adjustment of the real data and solving the general expression of Equation (5). The obtained kinetic parameters for each attribute were used for calculating the shelf life of the drink by using Equation (8), which was formerly used by Fikry, et al. [21]. Figure 2 presents the technique used to describe the shelf life of the drink by integrating the quality data.

$$\text{SL} = \frac{C - C_0}{k} \tag{8}$$

where SL is shelf life (day).

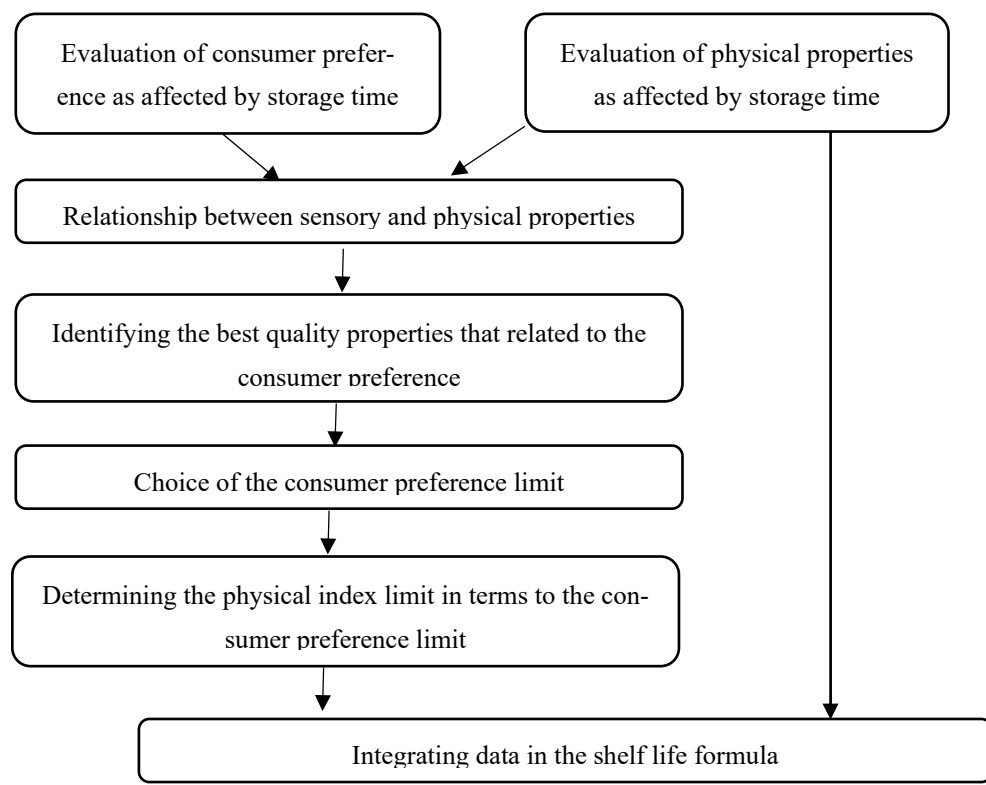

**Figure 2.** An approach used for forecasting the shelf life of a drink according to the quality indices.

*2.8. Data Analysis Protocol*

The collected data were exposed to one-way analysis of variance (ANOVA) with post hoc Tukey's test at $p < 0.05$ to explore the influence of the predictors on the responses (pH, colour, and sensory properties). Regression analysis technique was applied to measure the model quality for describing the actual data. The best model was chosen based on these criteria: the highest correlation coefficient, $R^2$ ($\geq 0.80$); the lowest percentage error, PE% (below 10%); and the lowest values of root mean square error, RMSE (the closest to zero) [22,29]. These parameters were estimated using the following formulas (Equations (9)–(11)). In order to analyse and model the data, the Minitab v. 18 statistical package (Minitab Inc., State College, PA, USA) was employed.

$$R^2 = 1 - \frac{\sum_{i=1}^{N}\left(y_{pred} - y_{exp}\right)^2}{\sum_{i=1}^{N}\left(y_{exp} - \overline{y_{exp}}\right)^2} \tag{9}$$

$$PE(\%) = \frac{100}{N} \sum_{i=1}^{N} \frac{\left|y_{exp} - y_{pred}\right|}{y_{exp}} \tag{10}$$

$$RMSE = \sqrt{\frac{\sum_{i=1}^{N}\left(y_{exp} - y_{pred}\right)^2}{N}} \tag{11}$$

where $y_{exp}$, $y_{pred}$ and $\overline{y_{exp}}$ refer to the experimental, predicted, and average values, respectively, and $N$ is the number of real data.

## 3. Results and Discussion

*3.1. Changes in EC, TSS, and Acidity of the Energy Drink and Their Kinetic Modelling*

EC is a measure of a material's ability to pass an electrical current. Table 1 represents the mean values of EC of the non-treated, sonicated, and thermally processed drink samples.

The average value of EC of the samples was 4.20 ± 0.005 mS/cm. The statistical results showed no significant differences in EC values among non-treated, sonicated, and thermally processed drink samples. It has been suggested that EC reflects the presence of ions and dipole components in the sample [31]. Comparable results were found for strawberry, orange, apple, pear, and tomato juices [32].

**Table 1.** Means and standard deviations of TSS and EC of the control, sonicated and thermally processed drinks during different storage periods.

| Treatment | Storage Time (day) | TSS (Mean ± SD) °Brix | EC (Mean ± SD) (mS/cm) |
|---|---|---|---|
| Control | 0 | 16.60 ± 0.26 [a] | 4.15 ± 0.006 [a] |
| | 3 | 16.57 ± 0.06 [a] | 4.16 ± 0.005 [a] |
| | 7 | 16.77 ± 0.12 [a] | 4.2 ± 0.0051 [a] |
| | 14 | 16.67 ± 0.21 [a] | 4.15 ± 0.0057 [a] |
| | 21 | 16.67 ± 0.21 [a] | 4.18 ± 0.0051 [a] |
| | 28 | 16.67 ± 0.20 [a] | 4.17 ± 0.0052[a] |
| Thermal | 0 | 16.37 ± 0.31 [a] | 4.26 ± 0.0051 [a] |
| | 3 | 16.73 ± 0.06 [a] | 4.26 ± 0.0053 [a] |
| | 7 | 16.60 ± 0.10 [a] | 4.28 ± 0.0056 [a] |
| | 14 | 16.63 ± 0.12 [a] | 4.23 ± 0.0052 [a] |
| | 21 | 16.63 ± 0.12 [a] | 4.25 ± 0.0057 [a] |
| | 28 | 16.63 ± 0.12 [a] | 4.24 ± 0.0058 [a] |
| US20 * | 0 | 16.37 ± 0.31 [a] | 4.16 ± 0.0052 [a] |
| | 3 | 16.77 ± 0.06 [a] | 4.17 ± 0.0056 [a] |
| | 7 | 16.8 ± 0.10 [a] | 4.16 ± 0.0058 [a] |
| | 14 | 16.83 ± 0.06 [a] | 4.19 ± 0.0056 [a] |
| | 21 | 16.77 ± 0.06 [a] | 4.23 ± 0.0053 [a] |
| | 28 | 16.80 ± 0.10 [a] | 4.20 ± 0.0055 [a] |
| US30 * | 0 | 17.30 ± 0.10 [a] | 4.29 ± 0.0053 [a] |
| | 3 | 17.37 ± 0.06 [a] | 4.26 ± 0.0052 [a] |
| | 7 | 17.27 ± 0.21 [a] | 4.26 ± 0.0054 [a] |
| | 14 | 17.4 ± 0.10 [a] | 4.23 ± 0.0054 [a] |
| | 21 | 17.23 ± 0.15 [a] | 4.19 ± 0.0055 [a] |
| | 28 | 17.27 ± 0.21 [a] | 4.21 ± 0.0053 [a] |
| US40 * | 0 | 16.37 ± 0.31 [a] | 4.16 ± 0.0057 [a] |
| | 3 | 16.4 ± 0.10 [a] | 4.17 ± 0.0058 [a] |
| | 7 | 16.3 ± 0.10 [a] | 4.16 ± 0.0053 [a] |
| | 14 | 16.4 ± 0.10 [a] | 4.15 ± 0.0054 [a] |
| | 21 | 16.27 ± 0.06 [a] | 4.18 ± 0.0053 [a] |
| | 28 | 16.3 ± 0.10 [a] | 4.16 ± 0.0054 [a] |

* US20, US30, and US40 refer to the sonication treatment times of 20, 30, and 40 min, respectively. [a] denotes no significant differences between different treatments.

The average TSS value of the drink samples was approximately 16.72 ± 0.14 °Brix (Table 1). Statistical analysis showed no significant differences between non-treated, sonicated, and thermally processed drink samples during the storage period. Some studies also reported that the TSS of juices such as strawberry [33,34] and orange, apple, pear, and

tomato juice [32] was not affected by different processes and the storage period such as with date juice [25].

It was found that pH plays a significant role in the efficient manufacturing, quality control, the growth of microorganisms and the shelf life of the products since it can directly affect the sensory characteristics of the drinks. Figure 3 indicates the changes in pH of the non-treated, sonicated, and thermally processed drinks during the storage period. Evidently, statistical results indicated that there were significant differences ($p < 0.05$) between the different drink samples (control, thermal treatment, and sonication treatments) in pH values. Additionally, there was a slow decrease in the pH of different samples as a function of the storage period. Similar findings for fruit juices were reported by Mosqueda-Melgar, et al. [32] and Shanta, et al. [25]. It was reported that the decline in the pH value could be due to the formation of acidic Maillard products and microbiological growth as well during storage [35,36]. This finding might be supported by the high correlation ($R^2 = 0.825$) between the pH and BI, which can be noticed in Figure 4. Excitingly, the acceptance limit of pH for beverages has been documented to be 4.8 since a beverage with a pH value higher than 4.8 can be categorized as a low-acid beverage [37]. Fruit drinks with low pH values (less than 5.0) are extremely disposed to attack thermo-acidophilic and acid-tolerant bacteria due to their low pH, high sugar content, and organic acids [32]. Hence, sonicated and thermally processed drinks could not be exposed to the attack of thermo-acidophilic and acid-tolerant bacteria during a storage period of 28 days compared with the control samples.

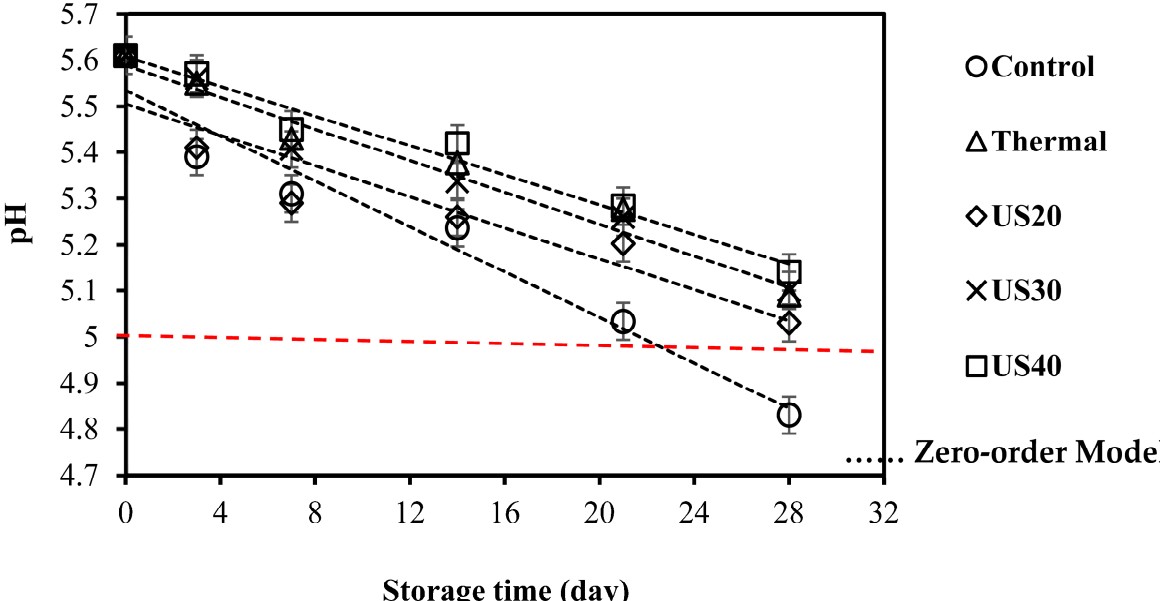

**Figure 3.** Changes in pH of control, sonicated for 20 min (US20), sonicated for 30 min (US30), sonicated for 40 min (US40), and thermally processed drinks during cold storage (the discrete red line refers to the safety limit of pH for beverages).

To model the changes in pH of non-treated, sonicated, and thermally processed drinks as a function of the storage period, the measured data of pH were fitted to the proposed models using the regression analysis technique. The statistical parameters for the proposed equations are listed in Table 2.

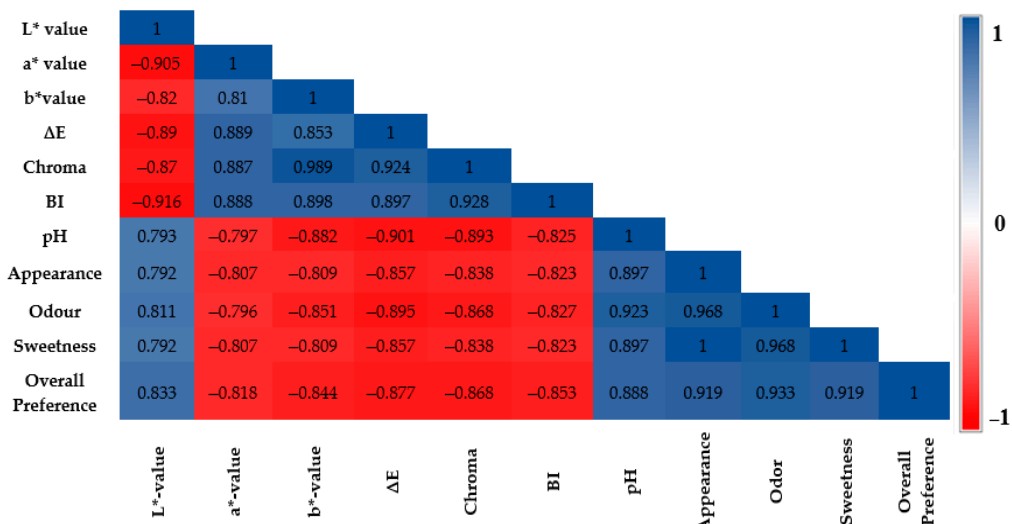

**Figure 4.** Correlogram of the physical and sensory properties of the energy drink from dates (note that the blue and red colour mean positive and negative relationships, respectively).

**Table 2.** Statistical parameters of zero-order kinetic model (the best model) that describes the changes in the properties of the energy drink.

| Treatment | Statistical Parameters | *L\** Value | *a\** Value | *b\** Value | *ΔE* | Chroma | BI | pH | Appearance | Odour | Sweetness | Overall Preference |
|---|---|---|---|---|---|---|---|---|---|---|---|---|
| Control | a | 47.67 | 12.27 | 29.97 | 3.24 | 32.39 | 279.19 | 5.53 | 8.34 | 7.90 | 8.342 | 8.14 |
| | b | −0.58 | 0.23 | 0.40 | 0.75 | 0.46 | 9.04 | −0.02 | −0.21 | −0.23 | −0.206 | −0.20 |
| | $R^2$ | 0.942 | 0.954 | 0.974 | 0.957 | 0.981 | 0.980 | 0.972 | 0.956 | 0.951 | 0.956 | 0.968 |
| | RMSE | 1.68 | 0.18 | 0.81 | 1.95 | 0.77 | 5.03 | 0.05 | 0.16 | 0.44 | 0.16 | 0.22 |
| | PE (%) | 3.82 | 1.03 | 2.00 | 9.50 | 1.73 | 0.93 | 0.86 | 2.22 | 7.99 | 2.22 | 4.11 |
| Thermal | a | 47.17 | 12.52 | 30.70 | 3.74 | 33.16 | 293.03 | 5.50 | 8.78 | 8.05 | 8.78 | 8.44 |
| | b | −0.390 | 0.175 | 0.294 | 0.51 | 0.339 | 5.154 | −0.011 | −0.132 | −0.146 | −0.132 | −0.150 |
| | $R^2$ | 0.862 | 0.894 | 0.951 | 0.901 | 0.953 | 0.932 | 0.844 | 0.922 | 0.931 | 0.980 | 0.890 |
| | RMSE | 1.92 | 0.39 | 0.48 | 2.04 | 0.58 | 9.34 | 9.34 | 0.29 | 0.32 | 0.29 | 0.46 |
| | PE (%) | 3.47 | 2.14 | 1.24 | 11.20 | 1.31 | 2.60 | 2.60 | 3.96 | 5.40 | 3.96 | 7.46 |
| US20 | a | 51.90 | 11.75 | 30.44 | 4.05 | 32.64 | 279.59 | 5.47 | 8.79 | 8.21 | 8.79 | 8.79 |
| | b | −0.46 | 0.18 | 0.22 | 0.54 | 0.27 | 3.63 | −0.01 | −0.19 | −0.21 | −0.19 | −0.19 |
| | $R^2$ | 0.858 | 0.890 | 0.938 | 0.869 | 0.947 | 0.974 | 0.801 | 0.985 | 0.958 | 0.955 | 0.949 |
| | RMSE | 2.57 | 0.38 | 0.37 | 2.54 | 0.32 | 6.68 | 0.10 | 0.13 | 0.27 | 0.13 | 0.28 |
| | PE (%) | 4.24 | 2.48 | 0.80 | 10.09 | 0.64 | 1.59 | 1.63 | 1.89 | 7.28 | 1.89 | 4.18 |
| US30 | a | 54.59 | 10.98 | 30.49 | 0.79 | 32.40 | 271.03 | 5.53 | 9.06 | 8.34 | 9.06 | 8.66 |
| | b | −0.45 | 0.18 | 0.19 | 0.50 | 0.25 | 3.00 | −0.01 | −0.13 | −0.14 | −0.13 | −0.11 |
| | $R^2$ | 0.934 | 0.860 | 0.853 | 0.966 | 0.908 | 0.946 | 0.843 | 0.894 | 0.920 | 0.894 | 0.904 |
| | RMSE | 0.52 | 0.46 | 0.81 | 0.64 | 0.60 | 2.60 | 0.48 | 0.33 | 0.25 | 0.33 | 0.21 |
| | PE (%) | 0.95 | 3.38 | 2.25 | 6.21 | 1.55 | 0.70 | 7.80 | 4.30 | 3.79 | 4.30 | 2.56 |

**Table 2.** *Cont.*

| Treatment | Statistical Parameters | *L* * Value | *a* * Value | *b* * Value | ΔE | Chroma | BI | pH | Appearance | Odour | Sweetness | Overall Preference |
|---|---|---|---|---|---|---|---|---|---|---|---|---|
| US40 | a | 52.68 | 10.45 | 30.83 | 2.26 | 32.57 | 279.17 | 5.61 | 8.98 | 8.26 | 8.98 | 9.05 |
| | b | −0.163 | 0.083 | 0.150 | 0.23 | 0.169 | 1.299 | −0.016 | −0.094 | −0.108 | −0.094 | −0.113 |
| | R² | 0.833 | 0.740 | 0.824 | 0.834 | 0.870 | 0.910 | 0.985 | 0.804 | 0.847 | 0.804 | 0.819 |
| | RMSE | 0.72 | 1.99 | 0.90 | 1.22 | 0.77 | 4.83 | 0.03 | 0.43 | 0.40 | 0.43 | 0.40 |
| | PE (%) | 1.04 | 14.43 | 2.35 | 11.57 | 1.89 | 1.32 | 0.39 | 4.83 | 5.43 | 4.83 | 4.34 |

* US20, US30, and US40 refer to the sonication treatment times of 20, 30, and 40 min, respectively.

The outcomes of the regression analysis revealed that the zero-order model displayed a high goodness-of-fit according to the high $R^2$, low PE%, and low RMSE values and can satisfactorily describe the experimental pH values.

*3.2. Changes in Colour Attributes of the Drink and Their Kinetic Modelling*

Colour attributes are considered the most critical quality characteristics as they contribute to consumer acceptance and control of the beverage industry processes [22,30]. Figure 5 depicts the experimental data of the colour properties (*L* *, *a* *, and *b* * values; ΔE; Chroma; and browning index) of the non-treated, sonicated, and thermally processed drinks as a function of storage times. The *L* * value is the supreme vital indicator of consumer preference. The results indicated that the *L* * value of the non-treated, sonicated (US20 and US30), and thermally processed drinks significantly declined with the storage time ($p < 0.05$) (Figure 5a). However, the storage period did not significantly affect the drink sonicated for 40 min (US40). The alteration in the *L* * value could be attributed to the browning and Maillard reaction. In contrast, Figure 5b–f indicates that the *a* * value, *b* * value, and Chroma of the non-treated, sonicated, and thermally processed drinks significantly increased with storage time. Although an increase in ΔE and BI was observed of the date drink sonicated for 40 min (US40), they were not significantly influenced by the storage period. The significant influence on the *L* *, *a* *, and *b* * values and the Chroma may be due to the non-enzymatic browning, which could influence the difference in the colour traits (*L* *, *a* *, and *b* * values) [38]. Formerly, it was reported that the non-enzymatic browning in stored food products could be affected by several factors, such as storage temperature, organic acids, $O_2$, and sugars [39]. These results are in agreement with those found by Kulkarni, et al. [4].

In order to model the changes in colour characteristics (*L* *, *a* *, and *b* * values; ΔE; Chroma; and BI) of non-treated, sonicated, and thermally processed drinks as a function of the storage period, the experimental values of the colour attributes (*L* *, *a* *, and *b* * values; ΔE; Chroma; and BI) of the drinks were fitted to the proposed models by using the regression analysis technique. Table 2 summarizes the statistical parameters for the proposed equations.

The outcomes of the regression analysis revealed that the zero-order model displayed a high goodness-of-fit according to the high $R^2$, low PE%, and low RMSE values and can satisfactorily describe the real values of the colour attributes (*L* *, *a* *, and *b* * values; ΔE; Chroma; and BI).

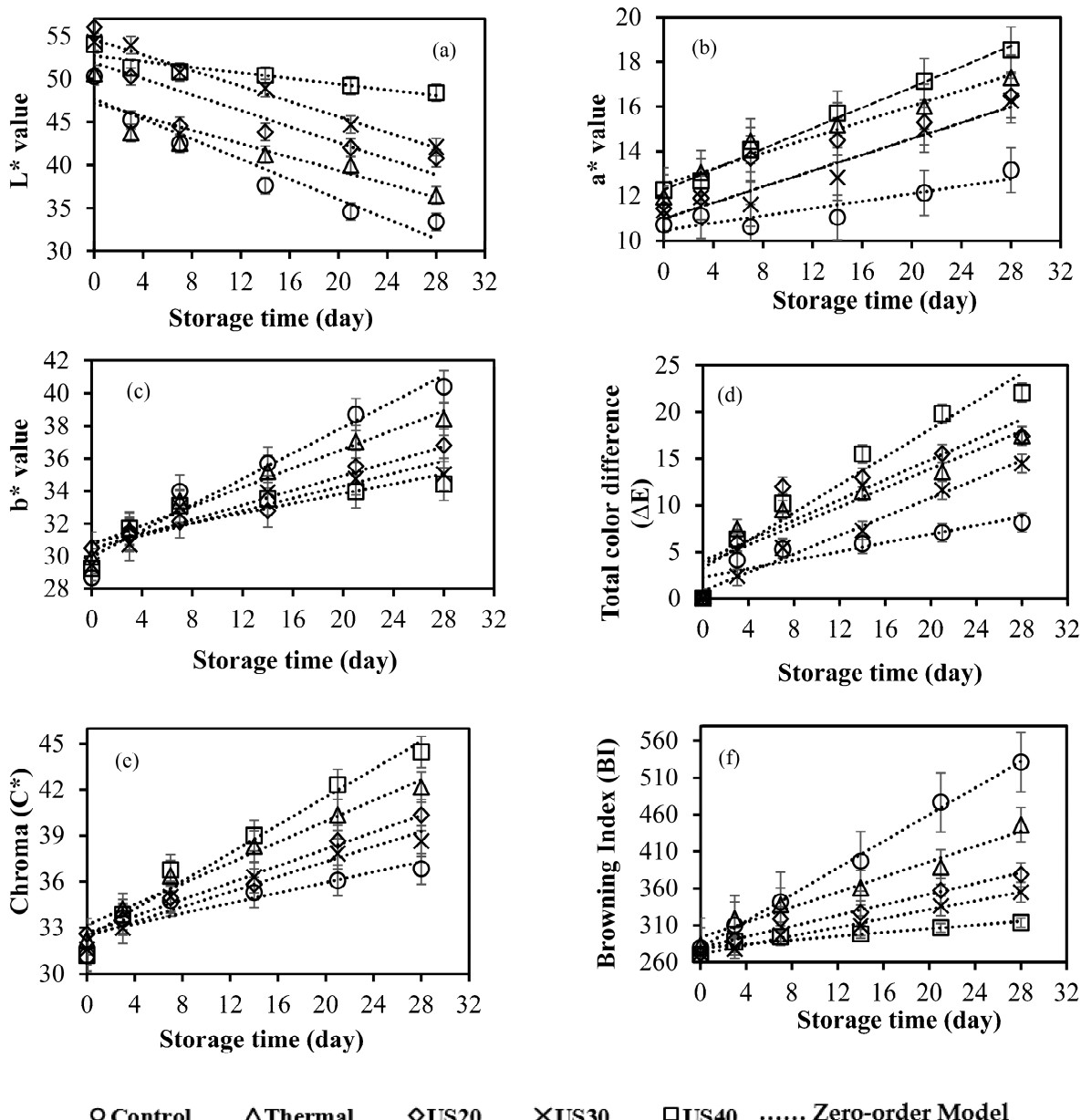

**Figure 5.** Changes in colour attributes, (**a**) *L\** value, (**b**) *a\** value, (**c**) *b\** value, (**d**) Δ*E*, (**e**) Chroma, and (**f**) BI of the control, sonicated for 20 min (US20), sonicated for 30 min (US30), sonicated for 40 min (US40), and thermally processed drinks during cold storage.

### 3.3. Changes in Sensory Properties of the Drink and Their Kinetic Modelling

Sensory evaluation is used for food quality control and product development. The sensory attributes of the control, sonicated, and thermally processed drinks at different storage times, such as the appearance, odour, sweetness, and overall preference, were investigated. A likely satisfactoriness limit can be nominated as a score equal to 6.0 on a nine-point hedonic scale [40]. Figure 6 demonstrates the changes in sensory attributes of the control, sonicated, and thermally processed drinks during cold storage periods.

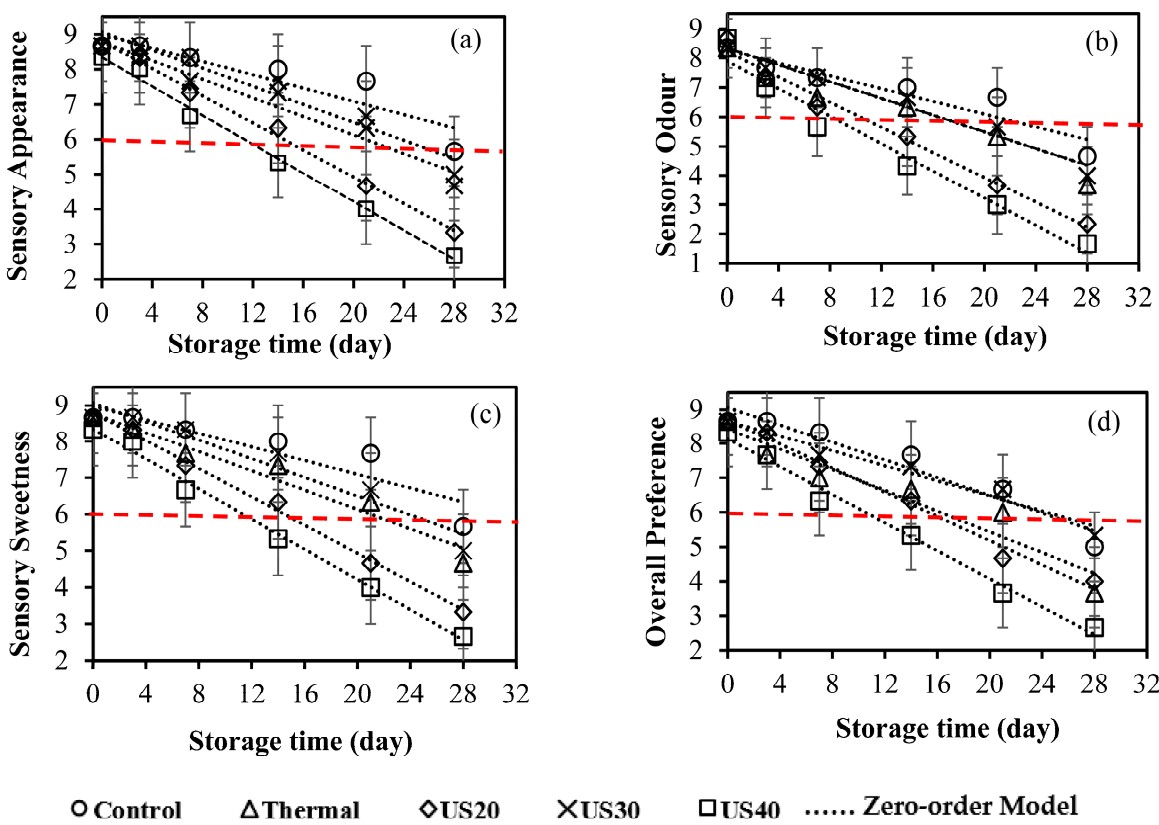

**Figure 6.** Changes in sensory attributes, (**a**) appearance, (**b**) odour, (**c**) sweetness, and (**d**) overall preference of sonicated for 20 min (US20), sonicated for 30 min (US30), sonicated for 40 min (US40), thermally processed and control drinks during cold storage.

Appearance is an influential factor that can indicate quality control during processing and storage [41–43]. According to Figure 6a, the processes and storage periods significantly affected the appearance of the control, sonicated, and thermally processed drinks. However, the appearance of the US40 sonicated samples kept desirable until 28 days compared to that for the control and US20 sonicated samples which degraded before 14 days of storage. In contrast, the appearance of both the US30 sonicated and thermally processed samples was still desirable until 21 days. A positive correlation between the L* value and the appearance of the drink is depicted in Figure 6. In contrast, the ΔE of the drink was negatively related to the appearance.

Odour is also considered a chief quality indicator of beverages. Figure 6b shows that the odour scores significantly decreased with the increased storage time. Nonetheless, the odour of the US40 sonicated samples remained desirable until 21 days of storage compared to that for the control and US20 sonicated samples which degraded before 10 days of storage. Meanwhile, the odour of the US30 sonicated samples was still desirable at 21 days compared to the odour of the thermally processed samples, which degraded at 21 days. The alteration in the odour acceptance was formerly justified by the decline in the pH of the drink, which affects the citric acid aroma [44]. It can be observed from Figure 4 that the correlation between the odour and pH of the drink was found to be positive ($R^2 = 0.923$).

Regards the sweetness attribute, it is clear from Figure 6c that the sweetness scores significantly decreased as the storage time increased. The decrease in the sweetness scores could be because of the drop in the pH of the drink, which influences the acid flavour [44]. Nevertheless, the sweetness of the US40 sonicated samples stayed desirable until 21 days compared to that for the control and US20 sonicated samples which degraded before 16 days of storage. Meanwhile, the sweetness of the US30 sonicated samples was still desirable

until 21 days compared to the sweetness of the thermally processed samples. A positive correlation between the pH and the sweetness of the drink is shown in Figure 4.

The overall preference scores of the drinks decreased with the increase in storage time (Figure 6d) as a result of decreased sensory parameters (appearance, odour, sweetness). However, the overall preference of the sonicated (US40, US30) and thermally processed samples kept desirable until 21 days compared to that for the control and US20 sonicated samples which were still accepted until 12 and 16 days of storage, respectively. Similar trends were reported for date juice during storage [4,25].

In order to model the changes in the sensory attributes (appearance, odour, sweetness, and overall preference) of non-treated, sonicated, and thermally processed drinks as a function of the storage period, the experimental values of the sensory attributes (appearance, odour, sweetness, and overall preference) of the drink were fitted to the proposed models by using the regression analysis technique. The statistical parameters for the proposed equations are illustrated in Table 2.

The outcomes of the regression analysis revealed that the zero-order model displayed the best fit according to the high $R^2$, low PE%, and low RMSE and can satisfactorily describe the real values of the sensory attributes (appearance, odour, sweetness, and overall preference).

### 3.4. Prediction of Shelf Life of the Energy Drink from Dates

The shelf life of a beverage can be defined as the interval needed to target the satisfactoriness limit or the period before quality properties degrade beyond a borderline [24]. Throughout the storage period, alterations in the quality attributes occur, resulting in degradation in the quality attributes, which then might limit aspects for the shelf life of the beverage. The constant change in quality attributes resulting from the kinetic model (Equation (5)) were employed in Equation (8) in order to predict the shelf life of the energy drink. The shelf life of the drink according to the pH, colour, and sensory attributes can be predicted using Equation (8). Table 3 shows that the predicted shelf life of the sonicated drink (US40) calculated according to its colour change and pH is longer than that for the control and thermally processed drinks. Meanwhile, the predicted shelf life of the US30 sonicated drink calculated according to overall appearance is longer than that for the control, thermally processed, and US20 sonicated drinks. Based on the three chosen criteria, we find that the overall appearance is the main controlling factor, which gave the greatest loss of estimated shelf life among all indicating properties.

**Table 3.** Predicted shelf life of drinks by integration of quality properties.

| Properties | Shelf Life (Day) | | | | |
|---|---|---|---|---|---|
| | **Control** | **Thermal** | **US20** | **US30** | **US40** |
| Colour change ($\Delta E$) | 13.6 | 26.7 | 23.9 | 23.3 | 30.4 |
| pH | 12.2 | 19.5 | 20.8 | 20.4 | 20.4 |
| Overall appearance | 9.8 | 17.8 | 13.3 | 18.3 | 15.6 |

### 4. Conclusions

In conclusion, this study investigated the effects of ultrasound and thermal treatments on the physical and sensory properties of an energy drink made from dates during cold storage. The results revealed significant differences in pH; $L^*$, $a^*$, and $b^*$ values; Chroma; BI; and sensory attributes ($p < 0.05$). During the storage period, the pH of the sonicated and thermally processed drinks gradually decreased, but they were still classified as low-acid beverages (pH > 4.5). The results indicated that the $L^*$ value of non-treated, sonicated (US20 and US30), and thermally processed drinks significantly decreased during the storage time. However, the storage period did not significantly affect the drink sonicated for 40 min (US40). The $a^*$ value, $b^*$ value, and Chroma of the non-treated, sonicated,

and thermally processed drinks significantly increased with the storage time. Although an increase in ΔE and BI was observed of the date drink sonicated for 40 min (US40), they were not significantly influenced by the storage period. The processes and storage period significantly affected the sensory properties of the control, sonicated, and thermally processed drinks. The sweetness scores significantly decreased as the storage time increased. The appearance of the US40 sonicated samples remained desirable until 28 days compared to that for the control and US20 sonicated samples. Meanwhile, the odour and sweetness of the US40 sonicated samples, the sweetness of the thermally processed samples, the appearance of both the US30 sonicated and thermally processed samples, the odour of the US30 sonicated samples, and the overall preference of the sonicated (US40, US30) and thermally processed samples were acceptable until 21 days. A positive correlation between the pH and the sweetness of the drink was shown. Moreover, a positive correlation between the $L^*$ value and the appearance of the drink was displayed. In contrast, the $\Delta E$ of the drink was negatively related to the appearance. The outcomes of the regression analysis revealed that the zero-order model was the best model that could satisfactorily describe the real values of pH, colour attributes ($L^*$, $a^*$, and $b^*$ values; $\Delta E$; Chroma; and BI), and sensory properties (appearance, odour, sweetness, and overall preference). The shelf life of the drinks according to the pH, colour, and sensory attributes was predicted. The predicted shelf life of the US40 sonicated drink calculated according to the colour change and pH is longer than that for the control and thermally processed drinks. Overall, this study provides valuable information for beverage manufacturers to control the quality properties of the energy drink from dates during processing and storage. Moreover, the study can be used as a basis for further research on the effects of ultrasound treatments on the phenolic compounds as well as the rheological, antioxidant, and microbiological properties of date-based energy drinks packed in different packaging materials under different storage conditions.

**Author Contributions:** M.F. constructed the experiments, visualized, and analysed the data and wrote the draft. Y.A.Y. and A.M.A.-A. conceptualized, edited, and revised the manuscript. S.M., H.A.S., S.A.-G. and N.N.A.K.S. edited and revised the manuscript. S.A.H.M.B., N.S.M.G. and K.K. prepared the methodology and revised the results. All authors have read and agreed to the published version of the manuscript.

**Funding:** This work was funded by the Prince Faisal bin Fahad Awards for Sports Research, administered by the Leader Development Institute under the Ministry of Sport in Saudi Arabia. The content is solely the responsibility of the authors and does not necessarily represent the official views of the Leader Development Institute under the Ministry of Sport in Saudi Arabia.

**Data Availability Statement:** Data is available upon reasonable request.

**Acknowledgments:** The authors acknowledge the grant of the Prince Faisal bin Fahd Awards for Sports Research 2021 from the Ministry of Sport, Saudi Arabia for the research funding. The authors would like to express thanks to The Ministry of Higher Education of The Arab Republic of Egypt for providing a fully funded postdoctoral fellowship to M.F.

**Conflicts of Interest:** The authors declare no conflict of interest.

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
