# Peer review of "Assessment of Physical and Sensory Attributes of Date-Based Energy Drink Treated with Ultrasonication: Modelling Changes during Storage and Predicting Shelf Life"

_processes, doi:10.3390/pr11051399_

Round 1

Reviewer 1 Report

The present study aims at assessing the effects on the sensory and physical attributes of date-based energy drinks when treated with ultrasonification processes in order to test the shelf-life. The present study is very interesting and it compares traditional thermal treatments with ultrasonication as a novel treatment in order to assess the effects on processed drinks over time. In order to give an in-depth view of this topic, the authors analyzed the pH, total soluble solids and electrical conductivity as physical parameters and several sensory attributes as well. The analysis performed give a good overview of the materials studied.

I believe that the paper is well written overall and the methods are carefully described. The objectives and the design of the study are clearly stated and identified in the Introduction section. The authors provided the interpretation of the obtained results in a well-structured manner and included the appropriate statistical analysis.

Moreover, given the depth of the analysis, the current study is giving a clear view of the differences between the thermal processes and eco-friend and sustainable processes like ultrasonification. Given the multitude of analysis performed, I believe the topic of the present paper is relevant and of interest for the readers of the journal.

 On the other hand, the conclusion part could be improved, maybe include some statistical findings to highlight the differences and the advantages of the conservation processes used. Also, the Introduction section can be improved by adding references regarding the current findings about the effect of ultrasonification on ther types of food products, compared to tranditional techniques.

Author Response

RESPONSE TO REVIEWER #1’s COMMENTS:

General Comment: “The present study aims at assessing the effects on the sensory and physical attributes of date-based energy drinks when treated with ultrasonification processes in order to test the shelf-life. The present study is very interesting and it compares traditional thermal treatments with ultrasonication as a novel treatment in order to assess the effects on processed drinks over time. In order to give an in-depth view of this topic, the authors analyzed the pH, total soluble solids and electrical conductivity as physical parameters and several sensory attributes as well. The analysis performed give a good overview of the materials studied.

I believe that the paper is well written overall and the methods are carefully described. The objectives and the design of the study are clearly stated and identified in the Introduction section. The authors provided the interpretation of the obtained results in a well-structured manner and included the appropriate statistical analysis.

Moreover, given the depth of the analysis, the current study is giving a clear view of the differences between the thermal processes and eco-friend and sustainable processes like ultrasonification. Given the multitude of analysis performed, I believe the topic of the present paper is relevant and of interest for the readers of the journal.

Appreciation: We very much appreciate your kind encouragement and useful comments that help us to continue our research work.

Comment #1: On the other hand, the conclusion part could be improved, maybe include some statistical findings to highlight the differences and the advantages of the conservation processes used. Also, the Introduction section can be improved by adding references regarding the current findings about the effect of ultrasonification on ther types of food products, compared to tranditional techniques.”

Response #1:

The introduction has been improved as per your request. This references added to the manuscript and was green highlighted.

Moreover, the conclusion has been modified. This modification could be observed in the manuscript and was blue highlighted.

Reviewer 2 Report

The study entitled “Assessment of Physical and Sensory Attributes of Date-based Energy Drink Treated with Ultrasonication: Modeling Changes During Storage and Predicting Shelf Life” is a very interesting one, I read the manuscript with pleasure, being clear and easy to navigate.. However, there are some small changes to be made:

L106: twice “the” word, delete one of it

L149: please write The with low case

L153: please write the reference Abid et al. in square brackets in the form of numbers, as required by the format of the journal.

L157: Same thing to Kortei et al.

L175: Please correct Fikry et al.

L188: Write the author who previously used this method.

L225-227: Write just the first author, then et al. – Mosqueda-Melgar et al.

L252: properties not proprieties

L267: Write the first author, then et al.

L341: Attributes with low case

L348, 349: critira, conroling??

Please add in the Conclusion section if the ultrasound method is adequate to be used in the beverage industry and what are your future perspectives.

Author Response

RESPONSE TO REVIEWER #2’s COMMENTS:

General Comment: “The study entitled “Assessment of Physical and Sensory Attributes of Date-based Energy Drink Treated with Ultrasonication: Modeling Changes During Storage and Predicting Shelf Life” is a very interesting one, I read the manuscript with pleasure, being clear and easy to navigate. However, there are some small changes to be made:

Appreciation: We very much appreciate your kind encouragement and useful comments that help us to continue our research work.

Comment #1: L106: twice “the” word, delete one of it

Response #1: Sorry, it was a typing error. However, one of them has been deleted and was yellow highlighted.

Comment #2:L149: please write The with low case

Response #2: Sorry, it was a typing error. However, it was corrected and was yellow highlighted.

Comment #3:L153: please write the reference Abid et al. in square brackets in the form of numbers, as required by the format of the journal.

Response #3: Thanks for this observation. The references were modified as per journal style. The modifications can be seen in the lines () and were yellow highlighted.

Comment #4:L157: Same thing to Kortei et al.

Response #4: Thanks for this observation. The references were modified as per journal style. The modifications can be seen in the lines () and were yellow highlighted.

Comment #5:L175: Please correct Fikry et al.

Response #5: Thanks. It was improved and was yellow highlighted.

Comment #6: L188: Write the author who previously used this method.

Response #6: It was stated that Equation 8 was formerly used by Fikry, et al. [16]. This sentence could be seen in line () and was yellow highlighted.

Comment #7:L225-227: Write just the first author, then et al. – Mosqueda-Melgar et al.

Response #7: Thanks for this observation. The references were modified as per journal style. The modifications can be seen in the lines () and were yellow highlighted.

Comment #8:L252: properties not proprieties

Response #8: Sorry, it was a typing error. However, it was corrected and was yellow highlighted.

Comment #9:L267: Write the first author, then et al.

Response #9: Thanks for this observation. The references were modified as per journal style. The modifications can be seen in the lines () and were yellow highlighted.

Comment #10:L341: Attributes with low case

Response #10: Sorry, it was a typing error. However, it was corrected and was yellow highlighted.

Comment #11:L348, 349: critira, conroling??

Response #11: Sorry, it was a typing error. However, it was corrected and was yellow highlighted.

 Comment #11: Please add in the Conclusion section if the ultrasound method is adequate to be used in the beverage industry and what are your future perspectives.

Response #12: The conclusion has been improved and the future perspectives were included. These modifications could be seen in the red lines.

Reviewer 3 Report

General comments

 The manuscript is well written in general and presents a very interesting and useful study. However, some points need to be improved before publication.

 Introduction

 1.       Please write the scientific name in italics.

2.       Line 45. Is there no more recent data? For the world's palm date production

3.       Line 46. “The date fruit is a good source of fiber” how much? Please specify approximate values for the above-mentioned contents

4.       how much in percentage of total production represents the third-grade dates that are usually discarded? It is important to understand the magnitude of the problem.

5.       Line 53. is it a by-product or a lower grade product?

6.       What another related research has been done? Authors should add specific information related to previous studies carried out with the fruit. Hasn't the same material been used before? whether heat-treated or non-heat-treated? specify previous results.

 Materials and Methods

 7.       Storage study. What material was used to store the control, sonicated, and thermally processed drink samples?

8.       Sensory analysis. Semi-trained 165 panellists. Under what standard are the panelists trained and under what standard was the sensory analysis performed?

Results and discussion

 9.       Table 1. Please use the same number for decimal places for the average and for the standard deviation. 16.60 ± 0.26

10.   Line 227. “It was reported that the decline in the 227 pH value could be due to the formation of acidic Maillard products” Since for the Maillard reaction to take place, an amino group and a carbonyl group are needed, how much protein does the raw material have?

11.   It is interesting to see the correlogram of the physical and sensory properties of the energy drink from dates. However, it is necessary to specify which of these relationships, whether positive or negative, are significant.  

12.  In the correlograms, the color intensity is proportional to the correlation coefficient, with darker boxes indicating stronger correlations. Are all correlations that have the same color intensity equally strong?

13.  What can you say about the shelf life compared to other products on the market?

Conclusions

 14.   It may be improved to highlight the scientific conclusion also and not only repeat the obtained results. Please add more information regarding the importance of the obtained results in the food industry and possibilities of further or complementary analysis.

Author Response

RESPONSE TO REVIEWER #2’s COMMENTS:

General Comment: “The manuscript is well written in general and presents a very interesting and useful study. However, some points need to be improved before publication.”

Appreciation: We very much appreciate your kind encouragement and useful comments that help us to continue our research work.

Comment #1:   Please write the scientific name in italics.

Response #1: Sorry, it was a typing error. However, it was modified and was green highlighted.

Comment #3:   Line 45. Is there no more recent data? For the world's palm date production

Response #2: Actually the data was obtained from FAOSTAT in which no data are available for 2021 and above.

Comment #3:    Line 46. “The date fruit is a good source of fiber” how much? Please specify approximate values for the above-mentioned contents

Response #3: approximate values of total dietary fiber has been added. It can be seen in the manuscript and were green highlighted.

Comment #4:    how much in percentage of total production represents the third-grade dates that are usually discarded? It is important to understand the magnitude of the problem.

Response #4: Actually, the third-garde dates quantity have not been determined. But, approximately 20% of the annual production of date fruits is lost during the post-harvest process, e.g., over-ripened date fruits, improper storage, handling, transportation, and contamination according to Oladzad, et al. [1].

Comment #5:   Line 53. is it a by-product or a lower grade product?

Response #5: It is a lower grade product but it is also can be considered as a byproduct because it is usually discarded in the palm date industry.

Comment #6:    What another related research has been done? Authors should add specific information related to previous studies carried out with the fruit. Hasn't the same material been used before? whether heat-treated or non-heat-treated? specify previous results.

Response #6: Yes, there are few studies which investigated the palm date juice under thermal conditions. Its findings were included in the manuscript and were yellow highlighted. However, no study modelled the changes in the quality properties under sonication and the shelflife of the palm date juice.

Comment #7:  Storage study. What material was used to store the control, sonicated, and thermally processed drink samples?

Response #7: Both non-treated and treated samples were filled in low density polyethylene bottles and then stored. This modification was added to the manuscript and was green highlighted.

Comment #8:   Sensory analysis. Semi-trained 165 panellists. Under what standard are the panelists trained and under what standard was the sensory analysis performed?

Response #8:

The panelists were trained and the sensory analysis were performed according to (ISO [International Organization for Standardization] 8589 1998).

Comment #9:   Table 1. Please use the same number for decimal places for the average and for the standard deviation. 16.60 ± 0.26

Response #9: Thanks. It was modified.

Comment #10:   Line 227. “It was reported that the decline in the 227 pH value could be due to the formation of acidic Maillard products” Since for the Maillard reaction to take place, an amino group and a carbonyl group are needed, how much protein does the raw material have?

Response #10: The protein in the raw material is around (2.3-5.6%).

Comment #11:   It is interesting to see the correlogram of the physical and sensory properties of the energy drink from dates. However, it is necessary to specify which of these relationships, whether positive or negative, are significant. 

Response #11: Actually, all of these relationships are significant. It was stated in the context.

Comment #12:   In the correlograms, the color intensity is proportional to the correlation coefficient, with darker boxes indicating stronger correlations. Are all correlations that have the same color intensity equally strong?

Response #12: The correlations which have the same color intensity equally strong because they lie in at the same range.  

Comment #13:  What can you say about the shelf life compared to other products on the market?

Response #13:  I think that the shelf life of the current product is similar to that in the markets.

Comment #14:   It may be improved to highlight the scientific conclusion also and not only repeat the obtained results. Please add more information regarding the importance of the obtained results in the food industry and possibilities of further or complementary analysis.

Response #14: The conclusion has been improved and the future perspectives were included. These modifications could be seen in the red lines.

  1. Oladzad, S.; Fallah, N.; Mahboubi, A.; Afsham, N.; Taherzadeh, M.J. Date fruit processing waste and approaches to its valorization: A review. Bioresource Technology 2021, 340, 125625, doi:https://doi.org/10.1016/j.biortech.2021.125625.
